# The Neurokinin-1 Receptor: A Promising Antitumor Target

Rafael Coveñas [1,2,*] , Francisco D. Rodríguez [2,3] and Miguel Muñoz [4]

1    Laboratory of Neuroanatomy of the Peptidergic Systems, Institute of Neurosciences of Castilla y León (INCYL), University of Salamanca, 37007 Salamanca, Spain

2    Group GIR-BMD (Bases Moleculares del Desarrollo), University of Salamanca, 37007 Salamanca, Spain

3    Department of Biochemistry and Molecular Biology, Faculty of Chemical Sciences, University of Salamanca, 37007 Salamanca, Spain

4    Research Laboratory on Neuropeptides (IBIS), Pediatric Intensive Care Unit, Virgen del Rocío University Hospital, 41012 Seville, Spain

*    Correspondence: covenas@usal.es; Tel.: +34-923294400

**Abstract:** The important role played by the substance P (SP)/neurokinin-1 receptor (NK-1R) system in cancer is reviewed: this includes tumor cell proliferation and migration, anti-apoptotic mechanisms, and angiogenesis. SP, through the NK-1R, behaves as a universal mitogen in cancer cells. The NK-1R is overexpressed in tumor cells and, in addition, affects the viability of cancer cells. NK-1R antagonists counteract all the previous actions mediated by SP through NK-1R. In a concentration-dependent manner, these antagonists promote tumor cell death by apoptosis. Therefore, NK-1R is a potential and promising therapeutic target for cancer treatment by using NK-1R antagonists (e.g., aprepitant) alone or in combination therapy with chemotherapy or radiotherapy.

**Keywords:** NK-1 receptor; substance P; cancer; aprepitant; NK-1 receptor antagonists

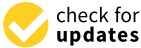

## 1. Introduction

The exponential increase in cancer research has, unfortunately, not been translated into better perspectives for cancer patients despite developing promising research fields focused on stem cells, new cytostatic compounds, and the human genome. Novel anticancer strategies for translational research must be explored and developed to fight this global chronic problem and to improve the diagnosis and treatment of tumors. Current cancer treatments are surgery, radiotherapy, and chemotherapy; however, the results are unsatisfactory in many cases. Unfortunately, cancer is still a major health problem worldwide (causing nearly 10 million deaths in 2020). Despite its severe side effects on essential organs, chemotherapy is currently the most used treatment against the disease. However, drug resistance appears in many cases, causing the failure of this antitumor strategy and leading to cancer cell invasion and metastasis [1]. Drug resistance often occurs after the fourth cycle of chemotherapy [2]. Hence, chemotherapy is not always an effective antitumor strategy, and the contribution of curative/adjuvant chemotherapy for 5-year survival in adult malignancies is approximately 2%. In addition, many patients with cancer suffer severe chemotherapy-associated side effects because the drugs used in chemotherapy do not specifically target tumor cells [3]. Consequently, chemotherapy affords a minor contribution to the survival of cancer patients, and although cytostatics are currently used in clinical practice, they are not future drugs. This means that new specific antitumor targets as well as strategies showing enhanced antitumor action and decreased toxicity must be urgently developed.

Tumor cells and the tissues they form suffer proliferation, survival, invasion, and metastasis (over 90% of cancer deaths are due to metastases and not to a primary tumor); these mechanisms are regulated by ligands (e.g., peptides) and their receptors, and, hence, one of the main goals in cancer research is to inhibit the cellular events mediated by these ligands [4]. Accordingly, it is a promising line of research to investigate in depth the roles

played by peptides and their receptors in cancer progression. Many peptides, including neurotensin, galanin, angiotensin II, endothelin-1, neuropeptide Y, cholecystokinin, gastrin, somatostatin, and calcitonin gene-related peptide, regulate cancer development [5–8]. It is widely known that the undecapeptide substance P (SP), through the neurokinin-1 receptor (NK-1R), is involved in chemotherapy-induced side effects (e.g., hepatotoxicity, neurotoxicity, nephrotoxicity, cardiotoxicity) and cancer development [1,9]. Moreover, the SP/NK-1R system has been related to poor prognosis, and this and other findings have opened up new research lines and antitumor therapeutic strategies [10–14]. Much published data has confirmed that NK-1R is a promising antitumor target (e.g., NK-1R is involved in the viability of tumor cells) [15,16]. Due to the vital role that the SP/NK-1R system plays in cancer, we update the involvement of NK-1R in cancer progression and the potential antitumor strategies targeting NK-1R.

## 2. Neurokinin-1 Receptor and Cancer

### 2.1. Neurokinin-1 Receptor: General Findings

NK-1R, also known as SP receptor or tachykinin 1 receptor, belongs to the rhodopsin-like G protein-coupled receptors family (seven-transmembrane domain receptors, serpentine receptors, or 7TM receptors). It is widely distributed by the whole body (e.g., nervous system, immune and endothelial cells, lung, gastrointestinal tract, skin), and it is highly conserved along the species [17,18]. The tachykinin receptor 1 (*TACR1*) gene with cytogenetic location 2p12 encodes the human NK-1R that expands through the plasma membrane, forming seven transmembrane domains and several intracellular and extracellular loops. The receptor's function may be altered by anomalous expression, heterodimerization with other receptors, point mutations, or derangement of post-translational modifications. Specific amino acid positions may suffer post-translational changes, including glycosylation, formation of a disulfide bridge, and lipidation modification (Figure 1). Hundreds of single nucleotide variants (SNVs) have been described [19], but their pathological significance has not been analyzed thoroughly. They ask for further analysis related to specific pathologies, including cancer. Different classes of G protein-coupled receptors may assemble in operative homodimers, heterodimers, and oligomers [20]. These macromolecular structures amplify and diversify signaling responses and, in some cases, may serve as specific therapeutic targets for several pathological entities [20,21]. No information on the physiological formation of NK-1R dimers or oligomers has been provided. NK-1R functions through functional monomers residing in membrane microdomains where cholesterol is essential [22]. However, the capacity to heterodimerize appears possible. Artificially built heterodimers of NK-1R and $\beta_2$-adrenergic receptors were functional and internalized as a complex [23]. Therefore, the formation of complexes with the intervention of NK-1R may have relevance for the homeostasis of NK-1R signaling in abnormal or adaptive circumstances and requires further exploration.

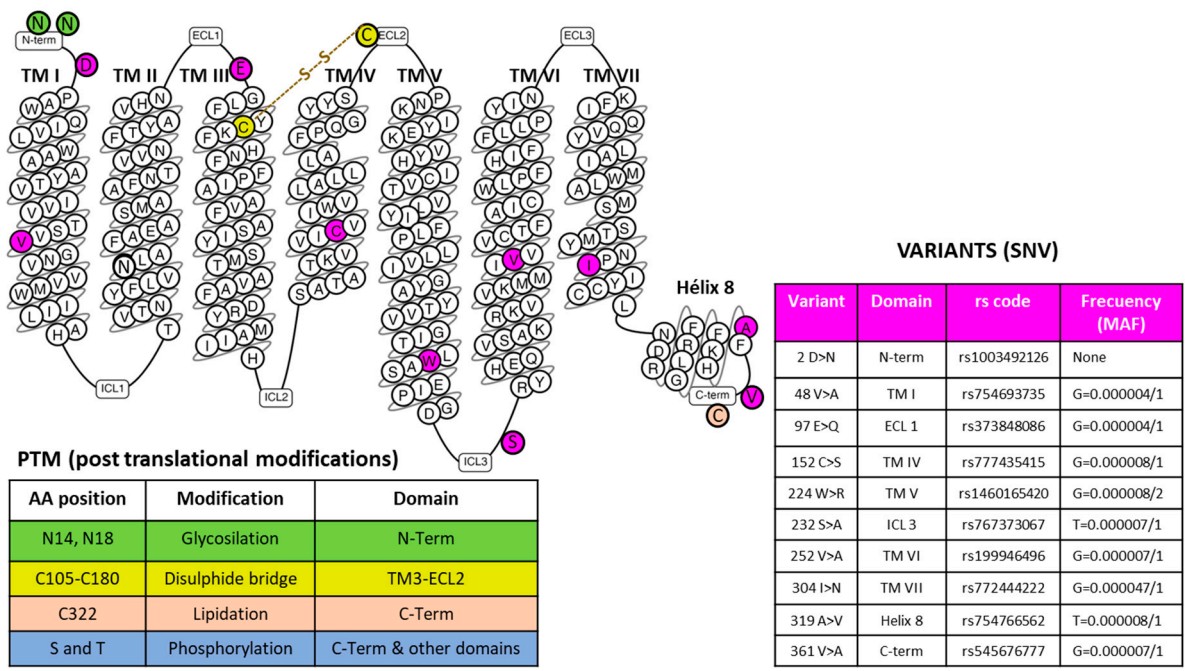

**Figure 1.** The snake plot of the human NK-1R (modified from [24]) highlights amino acid positions involved in post-translational modifications (PTM) and representative SNVs (Single Nucleotide Variations) in different receptor domains, as indicated. The table on the right-hand side of the figure showing the example variants depicts the amino acid position and change, the domain localization, the registered rs code, and the frequency acquired from the Single Nucleotide Polymorphism Database (dSNP) [19]. Abbreviations: ECL, extracellular loops; ICL, intracellular loops; MAF, minor allele frequency; TM, a transmembrane domain.

Seven-transmembrane helix receptors show a carboxy-terminal cytoplasmic domain (involved in the desensitization of the receptor), an amino-terminal extracellular domain (involved in the specificity of the receptor), and three intracellular and extracellular loops flanked by seven intermembrane domains [25]. The extracellular N-terminus and the intracellular C-terminus contain asparagine glycosylation and serine/threonine phosphorylation sites, respectively, which regulate NK-1R signaling [25]. NK-1R shows different active conformations, each of which has a different affinity for distinct agonists or antagonists: SP binds to the extracellular loops, and non-peptide NK-1R antagonists bind deep between the III and VI transmembrane segments of NK-1R [25]. Specific residues of this receptor (e.g., Gln165, His197, and 265) regulate the binding of non-peptide NK-1R antagonists [25]. NK-1R can be coupled to Gαq, Gαs, Gαi, Gαo, and Gα$_{12/13}$ proteins, the activation of a determined G protein being regulated by the type of ligands and NK-1R conformation [25–29]. G proteins differ in the effectors/signaling pathways they activate, and via these pathways, the transcription of specific genes is controlled [2]. Gαi blocks adenylate cyclase activity, decreases the level of cyclic adenosine monophosphate, and increases the phosphorylation of extracellular signal-regulated kinases. Gαs activates adenylate cyclase; Gαq promotes the synthesis of inositol triphosphate, activates phosphatidylinositol-3 kinase (PI3K), and increases the intracellular concentrations of Ca$^{++}$. Gαo activates the Wnt-β-catenin signaling pathway, and Gα$_{12/13}$ activates the Rho/Rock signaling pathway and regulates cytoskeletal rearrangements [25,30–33]. Through these mechanisms, SP, after binding to NK-1R, regulates the anti-apoptotic cell proliferation and cell migration signaling pathways involved in cancer development [34]. Figure 2 shows the most representative NK-1R downstream pathways involved in cancer progression.

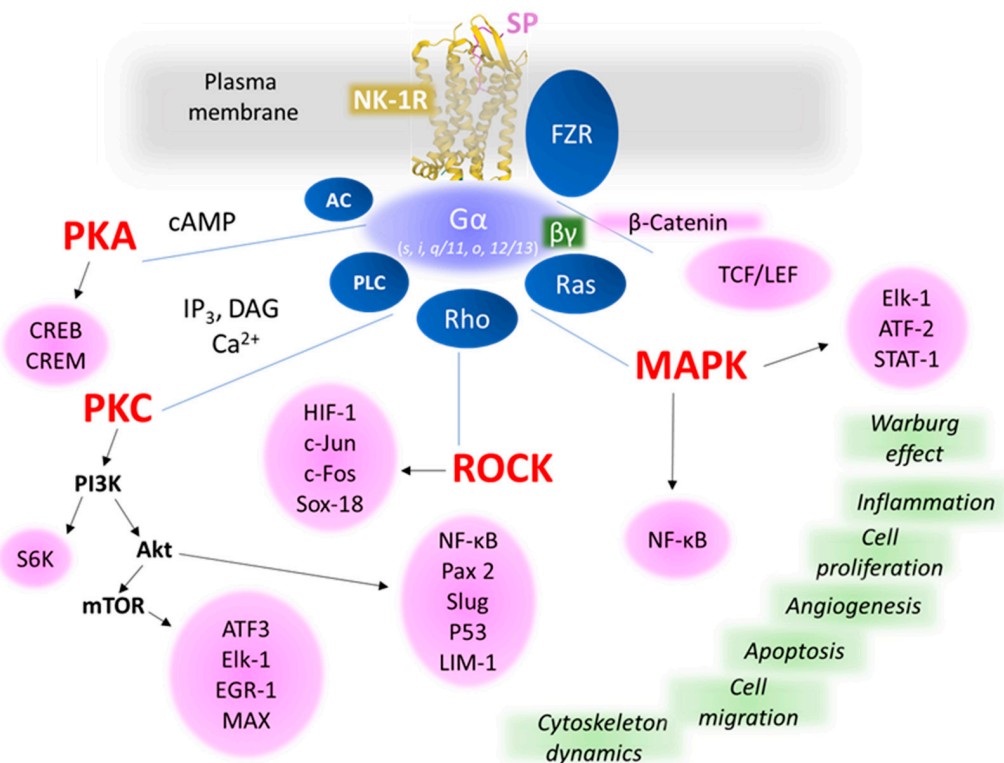

**Figure 2.** Representative NK-1R intracellular signaling pathways, which are possibly involved in cancer-associated processes. The receptor figure in yellow corresponding to PDB ID 7RMH [35] is from the Protein Data Bank [36] drawn with Mol* web-based open-source toolkit [37]. The transcription factors responsible for controlling cellular events (green) are in pink circles. Abbreviations: AC, adenylyl cyclase; Akt, Ak strain transforming, a protein kinase; ATF, activation transcription factor; cAMP, cyclic adenosine 5′monophosphate; CREB, cAMP response element-binding; CREM, cAMP responsive element modulator; c-Fos, transcription factor; c-Jun, transcription factor; DAG, diacylglycerol; EGR-1, early growth response; Elk-1, ETS-like protein; FZR, Frizzled receptor; HIF, hypoxia-Inducible Factor; IP3, inositol 1,4,5 trisphosphate; LIM-1, Lin11, Isl-1 y Mec-3; MAPK, mitogen-activated protein kinase; MAX, myc-associated factor X; mTOR, mammalian/mechanistic target of rapamycin, a protein kinase; NF-κB, nuclear factor kappa-light-chain-enhancer of activated B cells; p53, tumor suppressor protein; PAX-2, paired box gene 2; PI3K, phosphatidylinositol 3 kinase; PKA, protein kinase A; PKC, protein kinase C; PLC, phospholipase C; Ras, rat sarcoma virus, a small GTPase; Rho/ROCK, Ras homologous/Rho-associated protein kinase; Slug, SNAI2, a zinc-finger transcription factor; Sox18, SRY-related HMG-box; S6K, ribosomal protein S6 kinase; TCF/LEF, T cell factor/lymphoid enhancer factor family.

## 2.2. Substance P and Neurokinin-1 Receptor Antagonists

NK-1R shows a preferential affinity for SP and hemokinin-1 (a peripheral ligand of NK-1R), which belong to the tachykinin peptide family [17]. The affinity of the NK-1R for other members of the tachykinin peptide family, such as neurokinin A and B, is 100- and 500-fold lower than for SP. For this reason, NK-1R is also named the SP receptor [18]. SP and hemokinin-1 show a high and similar affinity for NK-1R. Both tachykinins share a homologous C-terminal sequence, favor angiogenesis, exert an anti-apoptotic effect, promote the proliferation and migration of tumor cells (e.g., hemokinin-1 increases the expression of matrix metalloproteinases, favors phosphorylation by protein kinase B and extracellular signal-regulated kinases (ERK), and enhances the action of the nuclear factor kappa-light-chain-enhancer of activated B cells) [18,38–40]. Accordingly, NK-1R antagonists inhibit the physiological actions mediated through the NK-1R by SP and hemokinin-1 [41]. SP

is an undecapeptide derived from the pre-protachykinin A gene (the *TAC1* gene locates on chromosome 7 in humans); it is hydrolyzed by p-endopeptidase (extracellular fluid) and angiotensin-converting enzyme (plasma), and it has a half-life from seconds/minutes (tissues) to hours (plasma) [25]. The undecapeptide is widely distributed by the whole body (e.g., immune and endothelial cells, smooth muscle, nervous system, fibroblasts, platelets, cerebrospinal fluid, blood) and, after binding to NK-1R, it is involved in many physiological and pathophysiological actions: micturition, chemotaxis of leukocytes, sensory perception, cardiovascular and respiratory mechanisms, salivation, movement control, immunological processes, sperm cell motility, platelet aggregation, cellular shape change, neuronal degeneration, memory, neurotransmitters release (e.g., acetylcholine, glutamate, histamine, dopamine), permeability of the blood–brain barrier, gastric motility, nausea and vomiting, obesity, pain, inflammation, anxiety, depression, bipolar disorder, pulpitis, neurocysticercosis, thrombosis, Hirschsprung's and Crohn's diseases, heart failure, myocarditis, cholestasis, emesis, migraine, mycosis, urinary incontinence, hepatitis, seizure, pruritus, dermatitis, acute pancreatitis, epilepsy, aggressive behavior, rheumatoid arthritis, asthma, chronic bronchitis, ulcerative colitis, viral and bacterial infection, alcohol addiction, and cancer [25,42,43]. The SP/NK-1R system is upregulated in many pathologies (e.g., inflammation, asthma, virus infection, acute pancreatitis, chronic stress, major depressive disorder, ulcerative colitis, Crohn's disease, and cancer) [25]. Patents using NK-1R antagonists against corneal neovascularization, ocular pain, melanogenesis, cough, bacterial infection, respiratory tract diseases, cardiomyopathy, pruritus, emesis, and cancer have been reported [43]. Despite the potential therapeutic actions of NK-1R antagonists to treat many human pathologies, its potential is currently minimized. It is important to remark that many of the beneficial actions of NK-1R antagonists seen in preclinical studies have been ineffective in clinical trials [42,44]. This observation could be due to a lack of knowledge regarding the molecular interactions between NK-1R and NK-1R antagonists, the non-appropriate selection of patients/endpoints in the clinical trials, the non-appropriate use of experimental animal models, and the lower dose of NK-1R antagonists administered. Thus, studies focused on new NK-1R antagonists showing improved pharmacokinetic characteristics are crucial, and must urgently be performed. In addition, for each pathology, the correct dose of the NK-1R antagonist must be administered because, for example, in clinical trials, aprepitant either exerted (300 mg/day for 45 days) or did not exert (160 mg/day for 5–6 days) an antidepressant action, depending on the dose administered [45,46]. Thus, the lower dose did not reach the threshold required to promote an antidepressant activity, which could be due to the number of NK-1Rs that must be blocked with aprepitant; it seems that the appropriate dose links to the number of NK-1Rs expressed in cancer cells.

### 2.3. Neurokinin-1 Receptor Isoforms and Cancer

The expression levels of NK-1R are elevated in cancerous samples obtained from the gallbladder [47], pancreas [48], metastatic breast cells [49], acute myeloid leukemia blasts [50], esophageal squamous cell carcinoma [51], or lung cancer cells [52,53]. In contrast, neighboring non-affected tissues showed regular expression. Two isoforms of NK-1R have been reported: full-length (407 amino acids) and truncated (311 amino acids, the last 96 residues at the C-terminus are lost) forms [54]. In some breast cancer cells, the complete structure of NK-1R diminished, and the truncated variant was more abundant [55]. The two variants may be involved in segregated expressions, responses to ligand concentrations, and triggering different intracellular signaling mechanisms [56]. The implication of NK-1R increased activity through both receptor forms in cancer cells is supported by the fact that specific antagonists reverse the system's participation in cell proliferation, migration, and metastasis [9]. Although precise mechanisms expect determination, there seems to be a dysregulated transcription, where different transcription factors may intervene. Indirect experimental evidence indicates that a putative regulator is a nuclear factor kappa B (NF-κB). This transcription factor governs the expression of NK-1R in macrophage cells by a process dependent on cytokine 1L-1β [57]. In addition, a pro-inflammatory cytokine

cocktail (interferon-γ, tumor necrosis factor-α, and 1L-1β) provoked overexpression of NK-1R in colonic epithelial cells and colonic epithelial cell lines [58]. However, the order in which molecular events occur inside the cells requires extensive analysis [56]. Still, the data support that this transcription factor plays a significant role in the hyperactivation of the NK-1R system related to cancer [59]. The short variant of NK-1R may control the immune environment that could suffer alterations at high concentrations of the ligand SP [60]. The oncogenic isoform of NK-1R is the truncated form that mediates tumor growth and malignancy. In contrast, the full-length form interacts with β-arrestin, which is involved in the desensitization, internalization, and endocytosis of NK-1R [18]. The expression of NK-1R mRNA is lower in benign tissues than in malignant ones. Compared with fibroblasts, the expression of the *TACR1* gene was augmented 7.5–30 times in human hepatoblastoma cell lines [48,61,62]. The truncated form is dominantly expressed in cervical and prostate cancer cell lines [11,63]. It seems that the truncated form prolongs the response of ligands, because its internalization and desensitization are affected (the absence of amino acid sequences at the C-terminus of NK-1R could block the internalization of the receptor, a clathrin-dependent mechanism, and the recycling processes, leading to SP longer response and cancer progression). In addition, human hepatoblastoma cell lines overexpress the truncated form. Still, negligible levels of this form were found in nonmalignant HEK-293 cells and human fibroblasts [61,64–66]. An increase in SP/NK-1R staining has been reported in metastatic tumors. In cultured normal epithelial cells, the level of SP was lower than that found in cultured cancer cells [62,67]. SP, via NK-1R, favors metastasis in human colorectal cancer cells [68]. The expression of NK-1R isoforms is important since activating the truncated form increases metastasis, whereas the activation of the full-length decreases it [55]. Moreover, the activation of the full-length isoform inhibited the proliferation of tumor cells, whereas the activation of the truncated form promoted its proliferation [55]. The tumor growth factor β regulates the expression of the truncated form, an action counteracted by the NK-1R antagonist aprepitant [69]. Pancreatic ductal adenocarcinoma cells mainly expressed the truncated form. Aprepitant exerted its highest antitumor action against them when tumor cells expressed higher levels of this isoform; in this study, the authors highlighted that NK-1R could be an important target in future personalized medicine [70]. Moreover, the latter study also reported that the expression of NK-1R was lower in pancreatic ductal adenocarcinoma tissues than in normal tissues. A better overall survival rate was observed in patients showing a high level of NK-1R [70]. These are unexpected and contradictory findings compared with most of the previously published results; these findings may occur exclusively in pancreatic ductal adenocarcinomas [9,70]. This must be elucidated in future studies. A review focused on the structural dynamics and signaling cascades (e.g., mitogen-activated protein kinases (MAPK), hairy and enhancer of split 1 (Hes1)) of NK-1R has recently been published [71]. For example, Hes 1, a transcriptional blocker of the Notch signaling pathway, reduced the growth suppression of tumor cells when NK-1R was downregulated [72].

### 2.4. The Substance P/Neurokinin-1 Receptor System and Cancer: Key Points

The SP/NK-1R system is involved in the molecular bases of many human pathologies, including cancer. This fact means that in-depth knowledge of this system is a crucial step toward better understanding and management of cancer. In the last few years, this knowledge has dramatically increased. Currently, the known key points regarding the involvement of the SP/NK-1R system in cancer are the following: (1) cancer cells express/overexpress NK-1R (e.g., 40,000–60,000 NK-1Rs per glioma cell); (2) NK-1R is involved in the viability of tumor cells (e.g., acute lymphoblastic leukemia, melanoma, lung cancer); (3) NK-1R is not involved in the viability of normal cells; (4) tumor cells express a higher level of truncated NK-1R than normal cells; (5) tumor cells express a lower level of full-length NK-1R than normal cells; (6) SP is not relevant for the viability of cancer cells; (7) SP, after binding to NK-1R (activates MAPK cascade), promotes the mitogenesis/migration of cancer cells (solid and non-solid tumors); (8) cancer cells synthesize and release SP;

(9) the undecapeptide acts through endocrine (from tumor mass), paracrine and autocrine mechanisms; (10) SP comes from multiple sources: cancer cells, immune cells placed in the tumor microenvironment, nerve terminals, and bloodstream; (11) SP increases the expression of NK-1R but not that of other tachykinin receptors (NK-2R and NK-3R); (12) SP exerts an anti-apoptotic effect (activating the basal kinase activity of the anti-apoptotic molecule protein kinase B (Akt)) and increases the glycolytic rate (Warburg effect) of tumor cells (which augment their metabolism due to the glucose obtained); (13) SP promotes the growth of blood vessels (angiogenesis, favoring the development of tumors by increasing the supply of blood; endothelial cells express NK-1R and SP), and (14) a higher serum SP level and a higher number of NK-1Rs have been observed in patients with cancer than in healthy individuals [1,9,15,25,73–86]. Moreover, the *TAC* and *TACR1* genes are expressed in primary stem cells derived from human placental cord blood and human stem cell lines. SP promoted the proliferation and migration of stem cells; in the latter cells, SP activated the MAPK cascade via NK-1R, and NK-1R antagonists exerted an antitumor activity against cancer stem cells [87–90].

### 2.5. The Substance P/Neurokinin-1 Receptor System as a Cancer Predictive Factor

The overexpression of NK-1R by cancer cells can be used as a prognostic biomarker. It could also facilitate a specific antitumor treatment using NK-1R antagonists (e.g., aprepitant, CP-96,345, L-733,060, SR-140,333, L-732,138). In addition, an increased serum SP level can be used as a predictive factor, indicating a high risk of developing cancer [9,15,91]. The higher level of NK-1R in tumor cells has been related to cancer stage, tumor-node metastasis, poor prognosis, larger tumor size, and higher metastatic and invasion potential [1,72,73,75,92,93]. The expression of NK-1R has been associated with poor prognosis and advanced clinical stages of lung cancer [14]. A poor prognosis is associated with a high expression of truncated NK-1 in breast cancer, and the expression of SP/NK-1R has been suggested as a predictor for colorectal cancer [69,94]. NK-1R has been suggested as a tumor biomarker in hepatoblastoma, independent of the tumor biology and clinical stage. In non-tumor controls, the expression of the truncated NK-1R was lower than that reported in children with hepatoblastoma [2,95]. Tumor size and lymph-node metastasis relate to the number of fibers containing SP [96–98]. In breast cancer, the overexpression of SP is associated with a negative prognostic value, the expression of NK-1R with a high Ki-67 index (the nuclear Ki-67 protein is related to proliferative mechanisms), and higher tumor grade [99,100]. The data suggest that NK-1R and SP may serve as predictive cancer factors.

### 2.6. The Neurokinin-1 Receptor Is Crucial for the Viability of Cancer Cells

Another significant point is that the expression of the *TACR1* gene is essential for the survival of tumor cells but not for the viability of normal cells; this means that NK-1R is a promising and specific therapeutic target for cancer treatment [15]. Because the stimulus mediated by SP is beneficial for the survival of cancer cells (e.g., proliferation, migration, anti-apoptotic effect, NK-1R synthesis increase), these cells overexpress NK-1R, ensuring SP binding. Blocking the stimulus with NK-1R antagonists or silencing the expression of the receptor, tumor cells suffer apoptotic mechanisms [78,101–103]. Antibodies against SP promoted apoptosis and decreased epidermal growth factor receptor (EGFR) phosphorylation, as well as the survival of tumor cells. An increase in EGFR expression is associated with an increased expression of SP and a worse prognosis [104,105]. When tumor cells do not receive the stimulus mediated by SP, several mechanisms occur: the synthesis of cell cycle proteins halts, the number of apoptotic cells and endothelial growth factor receptors increases, and the steady state of human epidermal growth factor receptor 2 (HER-2) and the MAPK signaling pathway decrease [104,105]. NK-1R overexpression could render cancer cells extremely dependent on the SP stimulus, which could counteract the death-signal pathways of tumor cells. This overexpression could neutralize such pathways because death signals can be overridden by the strong SP mitotic stimulus. When blocking NK-1R using NK-1R antagonists, the balance can favor apoptotic signals, leading to cellular

death. It has recently been reported that the absence of NK-1R in glioma cells promoted the death of these cells by both apoptotic and necrotic mechanisms. An irreversible lesion, derived from a non-physiological situation, favors the breakage of glioma cell membranes, promoting their death by necrotic mechanisms [15]. This observation is another vital point worth studying in depth. Altogether, these findings demonstrate the crucial importance of the activation of NK-1R by SP for the survival of cancer cells.

### 2.7. Neurokinin-1 Receptor and EGFR, Akt, and HER-2

The activation of NK-1R by SP promotes EGFR transactivation, facilitates the formation of EGFR complex, activates the extracellular signal-regulated kinase (ERK) 2 and MAPK pathway, and induces DNA synthesis (Figure 2) [106]. ERK can be translocated into the nucleus, promoting proliferation and exerting an anti-apoptotic action. By activating ERK1/2, SP induced the proliferation and migration of cancer cells, and β-arrestin increased the sensitivity of cancer cells to NK-1R antagonists [107,108]. EGFR inhibitors blocked DNA synthesis and ERK2 activation mediated by SP, and EGFR transactivation mediated the mitogenic action promoted by SP [106]. EGFR and c-Src interact, and an augmented activity of c-Src has been associated with cancer progression, as this interaction enhances mitogenic signaling pathways [109]. Src kinase inhibitors block SP-dependent ERK phosphorylation and suppress the growth of tumor cells [110]. NK-1R, via EGFR transactivation, promoted non-small cell lung cancer progression (cell proliferation and migration) and aprepitant increased the sensitivity of lung cancer cells to gefitinib or osimertinib [111]. SP, via PI3K, augments the activity of protein kinase B (Akt), suppressing apoptosis. The inhibition of PI3K increased apoptosis and decreased cellular proliferation in cancer cells [29,79,112–114]. Akt activation has been related to poor prognosis and cellular processes that avoid the death of cells, leading to drug resistance and decreasing the antitumor effect of aprepitant [2,115,116]. Moreover, SP activates HER-2, which promotes drug resistance and malignant progression [104,117].

### 2.8. Neurokinin-1 Receptor and the Warburg Effect

SP, through NK-1R, mediates the Warburg effect (in tumor cells, compared with normal cells, the glycolytic rate is 200 times higher). This effect is blocked with NK-1R antagonists; then, tumor cells die by starvation (anti-Warburg effect) because NK-1R is needed to obtain glucose (Figure 2) [78,118]. SP activates glycogen synthase kinase-3 (GSK-3β), a finding associated with cancer progression and poor prognosis; its inhibition blocked tumorigenesis, counteracted the Warburg effect, increased apoptosis, and restrained cell motility [119,120]. NK-1R antagonists block GSK-3β activity and increase glycogen synthesis, counteracting the Warburg effect [117].

### 2.9. Neurokinin-1 Receptor Isoforms Balance

Cancer cells are highly responsive to NK-1R antagonists when they express a high quantity of truncated NK-1R. The overexpression of this form has been associated with an enhanced malignant potential, and the truncated form promoted the malignant transformation of non-tumorigenic cells. Cancer cells express more truncated than full-length forms, and the expression of the full-length form is inversely related to the proliferation of tumor cells, invasiveness, and metastasis [53,55,72,80,121]. The truncated form facilitated the synthesis of cytokines favoring growth-promoting actions, and the overexpression of miR-206 by tumor cells favored the malignant phenotype of these cells by maintaining a low level of the full-length isoform [92,122]. Thus, the antitumor effect mediated by NK-1R antagonists is associated with the differential expression of NK-1R isoforms; this is a significant point [2,34]. Accordingly, knowing the total number of full-length and truncated NK-1R isoforms expressed in tumors is necessary for the in-depth understanding of the mechanisms by which the SP/NK-1R system is involved in cancer, as well as the antitumor actions mediated by NK-1R antagonists.

## 2.10. Neurokinin-1 Receptor and Inflammation

SP enhanced the inflammatory-mediated tumor signaling pathways and promoted the expression of genes that facilitate tumor growth, invasion, and metastasis in head and neck cancers; these mechanisms were blocked with the NK-1R antagonist L-703,606 [123]. The SP/NK-1R system has been involved in the transition from chronic inflammation of the neck and head mucosa to preneoplastic/neoplastic transformation and development [111]. SP activates pro-inflammatory transcriptions factors (e.g., NF-κB) that control the expression of cytokines; NF-κB promotes the synthesis and release of pro-inflammatory cytokines (e.g., tumor necrosis factor; interleukins 1, 6, and 12), and an NF-κB binding site has been located in the promoter of the *TACR1* gene [25,124]. SP increases the release of cytokines by monocytes and macrophages, and cancer cells synthesize interleukin 6, being that its level is related to an increased progression of tumors; in addition, this interleukin has been located in the tumor cyst and cerebrospinal fluids of patients with glioma [124–127]. Previous data highlight the crucial roles played by SP on inflammatory mechanisms and tumor microenvironment: SP acts as a pro-inflammatory agent, and crosstalk between immune and tumor cells occurs. Chronic inflammation is associated with an increased risk of cancer development and truncated NK-1R/protein levels were higher in colonic epithelial cells with high-grade dysplasia and carcinoma than in quiescent colitis [128,129].

## 2.11. Neurokinin-1 Receptor, Metastasis, and Angiogenesis

The SP/NK-1R system activates genes (*survivin*, *HIF1α*, *MMP 1,9*, *TNFα*) involved in poor prognosis, increased occurrence of metastasis, malignancy, and cancer cell proliferation, as well as those involved in the prevention of apoptosis in tumor cells and related to resistance to doxorubicin (e.g., *FOXM1* gene) [112,130–136]. After binding to the NK-1R, SP activated the mammalian target of rapamycin (mTOR) signaling cascade, promoting cancer cell growth and metastasis, and favored the synthesis of degradative enzymes (matrix metalloproteinases) promoting cancer cell migration, invasion, and metastasis (Figure 2). All of these mechanisms were inhibited with NK-1R antagonists [40,47,66,137,138]. The SP/NK-1R system is involved in the migration of human breast and prostate tumor cells, and NK-1R mediated the migration of pancreatic cancer cells by upregulating the expression matrix metalloproteinases (MMPs). These actions were inhibited by NK-1R antagonists, which means that these antagonists could prevent both cancer recurrence and metastasis [139,140]. In this sense, SP secreted from breast cancer cells promoted the migration of these cells across the blood-brain barrier by releasing angiopoietin-2 and tumor necrosis factor α, which impaired the barrier (e.g., by inducing changes in the localization of tight junction proteins such as occludin-1 and claudin-5) and favored the infiltration of tumor cells into the nervous tissue; NK-1R antagonists counteracted these processes [141,142]. Moreover, a combined anti-hepatocellular carcinoma therapy (glycyrrhetinic acid, modified sulfated-hyaluronic acid, doxorubicin) blocked metastasis and drug resistance targeting the SP-hepatic stellate cells-hepatocellular carcinoma axis by inhibiting the SP-induced activation of hepatic stellate cells [143]. The SP/NK-1R system also promoted angiogenesis and the migration of ovarian cancer cells by increasing the expression of MMP 2 and 9, vascular endothelial growth factor (VEGF), and vascular endothelial growth factor receptor (VEGFR). These effects, mediated by SP, were blocked with aprepitant [144]. Tumor growth requires the formation of new blood vessels, which provide nutrients and oxygen to tumor cells; hence, angiogenesis is crucial for tumor development [145–147]. SP, via NK-1R, induces the proliferation of endothelial cells (angiogenesis), which is blocked with NK-1R antagonists [61,76]. Moreover, cells placed in the tumor microenvironment (e.g., macrophages and fibroblasts) can release angiogenic agents that increase vascularization. Endothelial cells of tumor vessels overexpress EGFR (this does not occur in endothelial cells located in normal vessels), which mediates the proliferation of endothelial cells after the binding of EGF [146–149]. Previous mechanisms regulate blood vessels (structure and function) around and within the tumor and also increase tumor blood flow.

*2.12. Neurokinin-1 Receptor and Substance P Located in the Nucleus of Cancer Cells*

NK-1R and SP have been located in the nucleus and cytoplasm of tumor cells (e.g., glioma): SP and the full-length NK-1R form were mainly observed in the nucleus, and the truncated NK-1R form in the cytoplasm [15]. The physiological significances of these findings are currently unknown; it is possible that the undecapeptide, through binding to NK-1R, could exert an epigenetic action by regulating DNA expression via modulating transcription factors and proto-oncogenes involved in several processes, such as the cell cycle, cellular differentiation/transformation, and apoptosis [81,150]. These observations must be investigated and elucidated.

*2.13. Neurokinin-1 Receptor and Nerve Terminals*

It is important to note that SP is released from nerve terminals and promotes tumorigenesis after binding to NK-1R. Peripheral nerves are essential in controlling the tumor microenvironment by activating tumor cells and promoting invasion and metastasis [143]. SP, via NK-1R, promoted perineural invasion in pancreatic cancer; this invasion was associated with a poor prognosis and inhibited by the SF10A (TNFRSF10A)/NF-κB pathway mediated by LOC389641 [151].

*2.14. Substance P: Contradictory Findings*

Finally, although most studies have shown that SP is involved in the proliferation and migration of cancer cells, others indicate that SP does not exert a proliferative action and that the undecapeptide promotes an antimetastatic effect [49,152]. The causes of these contradictory findings are currently unknown. Still, it could be due to the expression/number of truncated/full-length NK-1R isoforms, the involvement of distinct G proteins which can trigger different effector/intracellular signaling pathways, or specific unknown characteristics of the tumor cell lines studied in these previous works [49,152]; this must be clarified in future studies.

## 3. Antitumor Strategies Targeting the Neurokinin-1 Receptor

Many human cancer cell lines express NK-1R; thus, this receptor has been observed in oral, colon, gastric, lung, breast, endometrial, and lung carcinomas, as well as in melanoma, osteosarcoma, retinoblastoma, hepatoblastoma, glioma, and neuroblastoma [17,78]. A common antitumor strategy can be applied using NK-1R antagonists, and NK-1R antagonists (e.g., aprepitant, SR-140,333, CP-96,345, L-732,138, L-733,060) do exert an antitumor effect since all of them promote apoptosis in tumor cells [77]. NK-1R antagonists decrease the number of PD-1-positive cells, and it seems that through this mechanism, these antagonists attenuate the suppression of the immune response. This is because cancer cells inhibit immune cells by expressing inhibitory molecules, such as PD-L1, on their surface, escaping from the control of the immune system [153]. This observation is a crucial point that must be investigated in depth. Moreover, NK-1R antagonists cross the blood-brain barrier and reduce the permeability of malignant tumor cells across it, preventing brain metastasis. Antagonists also increase the expression of apoptotic markers (e.g., annexin V, propidium iodine), mRNA expression of pro-apoptotic targets (p21, p73, Bad, Bax, Bid), and the level of pro-apoptotic proteins (Bam, Bim, PARP, caspases 3 and 9). These drugs cause a decrease in anti-apoptotic proteins (Bcl-2, Bcl-xL) and inhibit c-myc expression (the molecule by which p73 regulates the activity of NF-κB; c-myc is crucial for the progression through the S phase to G2/M of the cell cycle) [25,77,139,154,155].

Currently, there are more than 300 NK-1R antagonists, but only six have been approved for clinical practice: aprepitant (oral administration, Emend, MK-869, L-754,030); fosaprepitant dimeglumine (intravenous administration, a water-soluble prodrug of aprepitant which is converted to aprepitant by phosphatases, Ivemend); Cinvanti (aprepitant injectable solution); rolapitant (Varubi, oral administration); netupitant (oral administration), and fosnetupitant (a prodrug of netupitant, intravenous administration). These compounds are non-peptide NK-1R antagonists used for treating acute/delayed chemotherapy-induced

nausea and vomiting as well as postoperative nausea and vomiting, which are mediated by SP [84,156–158]. Which is the best NK-1R antagonist to be used as a potential antitumor drug in clinical practice? The best one is the drug aprepitant, because the safety, pharmacokinetics, contraindications, and metabolism of this drug have previously been studied and because many preclinical studies have demonstrated its broad-spectrum antitumoral action [9,78,82]. Aprepitant blocks the proliferation and migration of cancer cells, promotes apoptosis in these cells, and exerts anti-Warburg and anti-angiogenic effects [77]. Residual tumor cells can remain in the tumor resection margin, provoking tumor recurrence after surgical procedures. Thus, the administration of aprepitant before and after surgical procedures has been suggested to prevent recurrence and metastasis since aprepitant exerts antiproliferative and antimetastatic actions [68,77].

NK-1R antagonists are, in general, safe and well tolerated even at high doses; most of the adverse events induced by these compounds were mild or moderate (e.g., dehydration, irritability, constipation, somnolence, alopecia, dizziness, tinnitus, fatigue, headaches, vertigo, hiccupping, and pharyngitis) [45,84,159]. Aprepitant is a highly selective NK-1R antagonist that binds to human NK-1R, exerting antiemetic, antiviral, antipruritic, antitussive, and antitumor actions [77,160]. The use of aprepitant in nuclear medicine for diagnosing and treating tumors expressing NK-1R has been reported [159,161,162]. Because aprepitant is a poorly water-soluble drug, several strategies have been developed to increase its dissolution and solubility and minimize food effects. In order to increase exposure, aprepitant has been developed as a nanoparticle formulation, increasing its bioavailability 3–4 times [163,164]. The half-life of aprepitant is 9–13 h, and it is metabolized in the liver by cytochrome P450, family 3, subfamily A (CYP3A4) [77].

The broad-spectrum antitumor action of aprepitant has been demonstrated in both in vitro and in vivo experiments against hepatoblastoma, esophageal squamous cell carcinoma, colon carcinoma, glioblastoma, gastric carcinoma, acute lymphoblastic leukemia, laryngeal carcinoma, cholangiocarcinoma, melanoma, neuroblastoma, acute myeloid leukemia, endometrial carcinoma, osteosarcoma, pancreatic carcinoma, and retinoblastoma, as well as against breast, ovarian, rhabdoid, prostate, and lung human cancer cell lines [1,10,11,13,43,67,70,77,82,91,144,165–169]. In most cases, the maximum inhibition (100%) was observed after the administration of $\geq$ 70 μM aprepitant [82]. Aprepitant/fosaprepitant decreased the volume of tumors (e.g., human hepatoblastoma or human osteosarcoma xenografts) in experimental animal models [168,169]. Aprepitant promoted apoptosis in colorectal cancer cells and blocked colorectal cancer xenograft growth; in addition, in human colorectal cancer patients showing a high level of NK-1R, poor survival was observed [170]. Aprepitant promoted apoptotic mechanisms in cancer cells by increasing mitochondrial reactive oxygen species (a $Ca^{++}$ flux into the mitochondria occurs from the endoplasmic reticulum, impairing its function) and decreasing or attenuating the activation of mTOR signaling axis p70 S6 kinase phosphorylation, Akt/p53 pathway, and the expression of c-myc, VEGFR 1, MMP 2/9, and VEGF A [1,25,171]. Aprepitant also impairs the interaction of beta-catenin with Forkhead box M1, leading to the inhibition of the Wnt canonical pathway. Additionally, it increases membrane stabilization of β-catenin, blocks the G2/M-phase cell cycle, increases the sensitization of cancer cells to cytotoxic action, and activates the caspase-3-dependent apoptotic cascade via the suppression of the anti-apoptotic target genes of transcription factor NF-κB [25,90,137,172–174]. Aprepitant induced cell cycle arrest, but not apoptosis, in human pancreatic ductal adenocarcinoma cells [70]. This NK-1R antagonist promoted caspase-dependent apoptotic cell death and G2/M arrest via the PI3K/Akt/NF-κB axis in cancer stem-like esophageal squamous cell carcinoma spheres [175]. The Wnt signaling pathway is involved in cell survival, proliferation, and differentiation; aprepitant blocks this pathway [2,137]. SP activates the Wnt signaling pathway, which is involved in the anti-apoptotic effect mediated by SP, and increases mRNA/protein expressions of Wnt signaling molecules (e.g., c-myc, cyclin D1, β-catenin) [176]. Aprepitant decreases Wnt signaling in Wnt-dependent cancer stem cells, which is vital since cancer stem cells are involved in tumor resistance and relapse [2,177].

Overactivation of the NF-κB pathway decreased the antitumor effect of aprepitant, and the activation of NF-κB by SP augmented NK-1R expression [133]. This activation was suppressed when NK-1R was blocked [123]. Moreover, the migration of cancer cells was reduced by decreasing the activation of NF-κB. It is known that NF-κB upregulated the truncated NK-1R form and slightly increased the full-length form [47]. Aprepitant reduced the production of reactive oxygen species and increased the expression and activity of superoxide dismutase and catalase in glioblastoma cells by promoting antioxidant mechanisms; this NK-1R antagonist inhibited the oxidizing action by SP [178].

Moreover, aprepitant significantly counteracted the reduced expression and activity of the thioredoxin system (thioredoxin, thioredoxin reductase) mediated by SP in glioblastoma cells [179]. SP decreased the antioxidant capacity of glioblastoma cells, increased malondialdehyde and reactive oxygen species levels, and reduced the thiol concentration. In contrast, aprepitant reduced the levels of malondialdehyde and reactive oxygen species, and increased the number of antioxidant components of the redox system in these cells [180]. Thus, NK-1R regulates the balance between the oxidant/antioxidant components of the redox system in glioblastoma cells. However, the previous findings must be confirmed in future studies since they were only carried out in one cancer cell line, namely U87 glioblastoma. Aprepitant induced endoplasmic reticulum stress, which promoted $Ca^{++}$ release and the suppression of the pro-survival ERK-c-myc signaling and also increased the pro-apoptotic expression of C/EBP-homologous protein [170]. SP altered the expression of apoptosis-related genes (*Bcl-2* upregulation, *Bax* downregulation) and the levels of cell cycle regulators (cyclin B1upregulation, p21 downregulation) in a cervical cancer cell line; these mechanisms were reversed with aprepitant [63].

It is important to note that the SP/NK-1R system is not only involved in solid tumors but also in hematological malignancies (e.g., acute myeloid leukemia (AML), acute lymphoblastic leukemia (ALL)); hence, the use of NK-1R antagonists is also possible in these malignancies [10,130,181]. AML patients showed an upregulation of NK-1R mRNA expression compared with healthy controls. AML cell lines expressed NK-1R and SP, but this expression was not observed in healthy individuals. The truncated NK-1R form was more abundant than the full-length form in AML cell lines and aprepitant sensitized AML cells to chemotherapeutic drugs [10]. NK-1R is also involved in the viability of human ALL cell lines; these cells express mRNA for NK-1R. SP promoted the mitogenesis of ALL cells, NK-1R antagonists induced apoptosis in ALL cells, and the *TAC1* gene was overexpressed in ALL cell lines [181]. Moreover, the damage promoted by aprepitant was higher in cancer cells than in non-cancer cells (e.g., the $IC_{50}$ for lymphocytes was ten times higher than that for AML cells) and, significantly, aprepitant did not promote a proliferative inhibitory action against human normal hematopoietic cells [130].

Severe side effects and chemoresistance are induced by radiotherapy and chemotherapy. SP/NK-1R mediates the following pathways related to chemoresistance in cancer: PI3K/Akt/mTOR, Notch 1, and Raf/Mek/ERK [76,153,182]. Both chemotherapy and radiotherapy promote inflammatory mechanisms, which are mediated by the release of SP from nerve terminals after binding to NK-1R [77]. However, combining both therapies with aprepitant promoted tumor radio and chemo sensitization, counteracting the side effects exerted by chemotherapy/radiotherapy (e.g., cardiotoxicity, nephrotoxicity), and induced a synergic antitumor action [1,9,183–187]. Combined cytostatics and radiotherapy promote mucositis, the breakdown of the mucosal barrier, and a systemic infection exacerbated by neutropenia. However, NK-1R antagonists attenuate these mechanisms [188]. Compared with current therapies, aprepitant in combination therapy with cisplatin was a safer and more effective therapeutic option against triple-negative breast cancer cells. In addition, aprepitant prevented chemotherapy-associated cardiotoxicity in cancer [135,166]. The nephrotoxicity mediated by cisplatin was counteracted by the NK-1R antagonist GR-205,171, improving renal function [186]. The co-administration of cytostatic drugs and NK-1R antagonists promoted an antitumor synergic effect on cancer cells. Still, this effect was not observed in human fibroblasts.

Further, when fibroblasts were treated with NK-1R antagonists before exposure to cytostatic drugs, the cells were partially protected from cytostatics [165]. Aprepitant and ritonavir exerted a synergic antitumor action against human glioma cells [189], and beneficial effects were found when aprepitant and chemotherapeutic agents were co-administered (5-fluorouracil, arsenic trioxide, ritonavir, temozolomide, cytosine arabinoside, doxorubicin, or cisplatin) [1]. Thus, for example, aprepitant decreased the neurotoxicity, hepatotoxicity, and nephrotoxicity of cisplatin, augmented the antitumor effect (chemo sensitization) of this chemotherapeutic drug, and prevented the side effects induced by the anticancer agent erlotinib [1]. Aprepitant increased the efficacy of chemotherapy, both in vitro and in vivo, by augmenting the sensitivity and overcoming resistance to 5-fluorouracil in colorectal cells via the induction of endoplasmic reticulum stress and the blockade of the ERK-c-myc signaling [170]. Moreover, aprepitant decreased apoptosis, the synthesis of reactive oxygen species, and doxorubicin-induced decrease in cell viability in cardiomyocytes [183]. It is important to note that the therapeutic index of cytostatics is very low, but that of aprepitant is much higher [142].

Previous data demonstrate that aprepitant could be used alone or in combination therapy to treat cancer. In this sense, NK-1R has been suggested as a therapeutic target in pediatric rhabdoid tumors, one of the most aggressive tumors in early childhood, since aprepitant exerted a growth inhibitory and apoptotic action against this tumor [13]. The study's authors concluded that aprepitant could be a promising antitumor agent against rhabdoid tumors, either alone or in combination therapy (e.g., cytostatic cisplatin) [13]. Moreover, the combination therapy of aprepitant (1140 mg/day for 45 days, compassionate use) and palliative radiotherapy induced, six months after treatment, the complete disappearance of the tumor mass ($8 \times 7$ cm) in a patient with lung cancer. In addition, no severe side effects were reported [159]. The dose of aprepitant administered in the previous case report was much higher than that administered when used as an antiemetic (first day: 125 mg; second day: 80 mg; third day: 80 mg) [159]. Accordingly, a clinical trial using high doses of aprepitant to treat patients who have lung cancer has been carried out; however, the results of the study have not yet been published [61]. Moreover, patents focused on the antitumor action of NK-1R antagonists, including aprepitant, have been reported [190,191]. The administration of NK-1R antagonists (e.g., aprepitant, casopitant, netupitant), either alone or in combination with chemotherapeutic agents (e.g., tamoxifen, raloxifene), might prevent cancer development in individuals with an enhanced risk of developing cancer. The recommended dose of antitumor is at least 20 times higher than that used in clinical practice as an antiemetic. [190]. NK-1R antagonists (aprepitant, vestipitant, L-733,060, casopitant) can treat some types of cancers (e.g., breast carcinoma, osteosarcoma, glioma, lung cancer, melanoma, rhabdomyosarcoma). The antitumor doses suggested are in a range between 2.8 and 28 mg/kg/day [191]. Altogether, the data previously reported advise that, in addition to exerting an antitumor action, NK-1R antagonists (e.g., aprepitant) also counteract the non-desirable activities exerted by cytostatics and radiotherapy. Accordingly, the antiemetic aprepitant drug is an excellent candidate to be administered as an antitumor drug, and its repurposing is urgently needed [9]. This objective will be faster and easier for aprepitant than for less-investigated NK-1R antagonists. The repurposing of aprepitant as a new therapeutic strategy to overcome cancer resistance has recently been suggested; this is a crucial point since drug resistance is the cause of over 90% of deaths in those individuals treated with chemotherapy or new anticancer treatments [167,192].

## 4. Neurokinin-1 Receptor Antagonists: Limitations and Challenges

High doses of aprepitant (e.g., 80 mg/day for seven months; 2 mg/kg/day from 6–84 days; 375 mg/day for two weeks) were safe and well tolerated in humans [193,194]. However, administering aprepitant and vincristine augmented the risk of chemotherapy-induced peripheral neuropathy. Aprepitant is an inducer/inhibitor of the cytochrome P450 CYP3A4 family of enzymes. Therefore, it can increase plasma levels of chemotherapeutic agents (e.g., midazolam, docetaxel) and corticosteroids [77,195] metabolized by the afore-

mentioned enzyme system. Carcinogenic doses of aprepitant (125–2000 mg/kg/day) have been reported in rodents (e.g., hepatocellular adenomas, skin fibrosarcoma); however, the possible antitumor doses of aprepitant to be administered in humans in future clinical studies could be much lower: 20–40 mg/kg/day [183].

New NK-1R antagonists derived from carbohydrates have also been suggested to treat different cancers (melanoma, lung carcinoma, breast cancer) [196]. The design of NK-1R antagonists aims to obtain compounds with advantageous properties, such as efficacy, selectivity, bio-disponibility, lack of secondary or toxic effects, access to designated tissues, and manageable chemical properties. These properties make up the basis for drug synthesis, which also counts on the known characteristics of the destination binding pocket within the receptor structure [197]. One of the first potent non-peptide NK-1R antagonists reported (compound CP 96,345) has a quinuclidine structure with a benzhydryl group and a methoxy-benzylamine ring [198]. Refinements and functionalization of NK-1R antagonists (for example, the introduction of a morpholine group and fluorination of phenyl rings) led to a family of drugs represented by aprepitant, the officially approved drug for treating vomiting and nausea caused by chemotherapy treatments [198,199]. Recent determination of crystal structures of NK-1R bound to antagonists has provided valuable information concerning the atomic interactions established between the protein and the antagonists, as well as their capability to adapt to different receptor conformations. These studies procure valuable data in order to design compounds with specific characteristics related to their binding and specificity properties. For example, aprepitant and antagonists with similar structural features bind to a hydrophobic pocket of NK-1R (Figure 3). The lateral chain of tryptophan in position 261 (W261) and other amino acid lateral chains in the vicinity provide a stabilizing environment for the antagonist to adapt to variable receptor conformations [200].

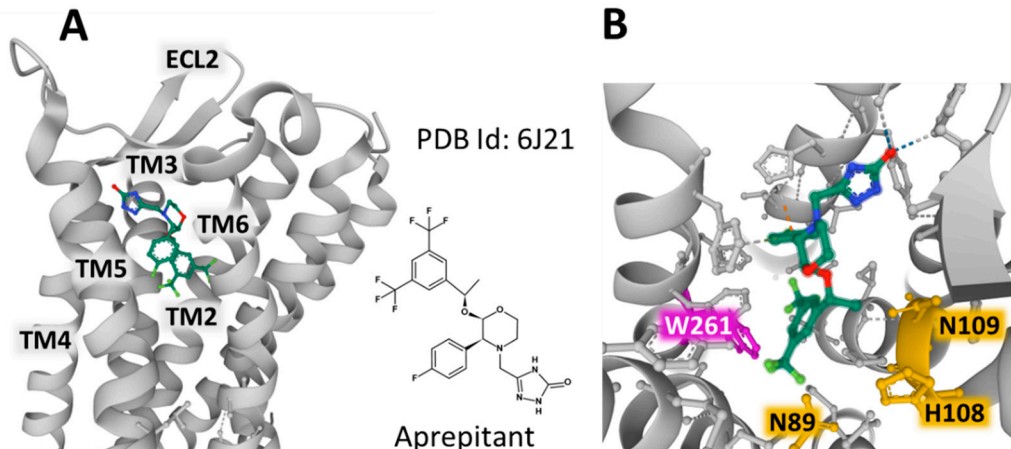

**Figure 3.** Panel (**A**) shows a general view of the structure of the antagonist aprepitant bound to NK-1R and its position in the environment provided by the transmembrane domains. Panel (**B**) depicts the structural relationship of aprepitant with key amino acid residues within the binding cleft. The figures in (**A**,**B**) are from the Protein Data Bank [36], PDB Id 6J21 [200], drawn with Mol* free web-based software [37]. The aprepitant 2D structure in the middle of the figure was illustrated using KingDraw free software [201].

The development of new antagonists should also consider the receptor's structure. The lateral chains of some amino acids allocated at the bottom of the antagonist binding cleft have amphiphilic or hydrophilic characteristics. Consequently, new antagonists with an added hydrophilic carbohydrate scaffold exhibited high binding activity and antagonistic effect by establishing weak interactions with amino acids N109, N89, and His 108 (Figure 3) [202]. On the other hand, the formation of NK-1R peptide antagonists bound to a lipid structure interacted with endosomes carrying internalized receptors and inhibited

intracellular NK-1R signals. Synthesis of variants of known antagonists that explore structural traits of the receptor at the molecular level may provide compounds that help to control NK-1R activity in a more specific and tuned fashion. The design and synthesis of new NK-1R antagonists which show different chemical structures to those currently known is an important line of research in medicinal chemistry and must be developed in the future. It is crucial to fully comprehend the function–structure relationships between NK-1R and SP for a rational design of new antitumor drugs.

## 5. Conclusions and Future Directions

The main findings regarding the involvement of NK-1R in cancer progression are summarized in Figure 4 and Table 1. SP exerts a proliferative action on cancer cells expressing NK-1R. The use of NK-1R antagonists emerges as a potential common specific antitumor treatment, irrespective of the tumor type.

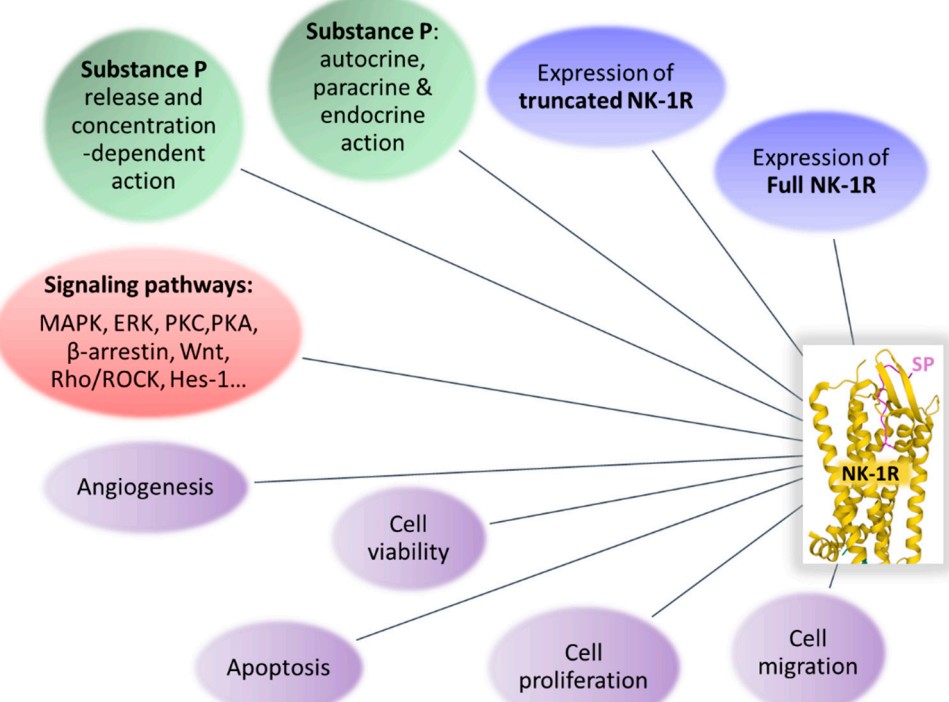

**Figure 4.** Neurokinin-1 receptors are involved in cancer initiation, progression, and metastasis in solid tumors and blood malignancies. Their action is conveyed through signaling mechanisms dependent on anomalous expression of the receptor isoforms and dysregulated action of substance P, locally or distantly released. The activities include control of cell intermediary metabolism, viability, proliferation, migration, angiogenesis, and apoptosis (see text for details). The receptor figure in yellow corresponding to PDB ID 7RMH [35] is from the Protein Data Bank (PDB), drawn with Mol* free web-based software [37].

**Table 1.** Involvement of NK-1R in cancer: main findings and antitumor strategies.

| |
|---|
| NK-1R: preferential affinity for SP/hemokinin-1 [17] |
| NK-1R: SP binds to extracellular loops; non-peptide NK-1R antagonists between III and VI transmembrane segments [25] |
| Gln165, His197, and 265 NK-1R residues: regulate the binding of non-peptide NK-1R antagonists [25] |
| NK-1R: coupled to Gαq, Gαs, Gαi, Gαo, and Gα$_{12/13}$ proteins [25–29] |
| NK-1R: regulates anti-apoptotic and tumor cell proliferation/migration signaling pathways and angiogenesis [34] |
| NK-1R: overexpressed in tumor cells [25] |
| Higher NK-1R level in tumor cells: related to cancer stage, tumor-node metastasis, poor prognosis, larger tumor size, and higher invasion/metastatic potential [1,72,73,75,92,93] |
| SP: increases NK-1R expression but not NK-2R/NK-3R expression [77,133] |
| NK-1R: involved in the viability of tumor cells [15] |
| Tumor cells: higher truncated NK-1R level and lower full-length form level than normal cells [61] |
| Full-length NK-1R: involved in NK-1R desensitization, internalization, and endocytosis [18] |
| Truncated NK-1R: oncogenic isoform mediating tumor growth and malignancy [18] |
| Activation of truncated NK-1R: increases metastasis and tumor cell proliferation; full-length activation decreases both mechanisms [55] |
| Patients with cancer: higher serum SP level and number of NK-1Rs [73,75] |
| NK-1R mRNA expression: lower in benign tissues than in malignant ones [48,61,62] |
| SP exerts a proliferative action on cancer cells expressing NK-1R. Many human cancer cell lines express NK-1R; a common antitumor strategy, irrespective of the tumor type, can be applied using NK-1R antagonists (aprepitant) [17,77,78] |
| Aprepitant: broad-spectrum antitumor agent [1,11,13,67,70,77,82,91,144,165,166,168,169] |
| Aprepitant: blocks proliferation/migration of cancer cells, promotes apoptosis, and exerts anti-Warburg/anti-angiogenic effects [77] |
| Aprepitant promotes apoptotic mechanisms by increasing mitochondrial reactive oxygen species [1,25,171]. |
| Aprepitant inhibits the Wnt canonical pathway, increases membrane stabilization of β-catenin, blocks the G2/M-phase cell cycle, increases the sensitization of cancer cells to cytotoxic action, and activates caspase-3-dependent apoptotic cascade [25,90,137,172–174] |
| NF-κB pathway overactivation: decreases the antitumor effect of aprepitant, and NF-κB activation by SP increases NK-1R expression; this activation is suppressed when NK-1R is blocked [123,133]. |
| Aprepitant exerted the highest antitumor action when tumor cells expressed higher truncated NK-1R levels [70] |
| NK-1R: involved in chemotherapy-induced side effects (hepatotoxicity, neurotoxicity, nephrotoxicity, and cardiotoxicity) [1,9] |
| Combination therapy of chemotherapy/radiotherapy with aprepitant: tumor radio and chemo sensitization counteract the side effects exerted by chemotherapy/radiotherapy and promotes a synergic antitumor action [1,9,183–187] |

Because tumor cells overexpress NK-1R, aprepitant is a broad-spectrum antineoplastic drug that acts in a concentration-dependent manner. However, aprepitant re-profiling is required to demonstrate its antitumor action and to translate preclinical findings to clinical practice. Both dose and days of administration without interruption must be increased compared to the current standard administration in clinical practice as an antiemetic (three days: 125, 80, and 80 mg). In order to exert an antitumor effect, aprepitant could be administered for a long period according to the treatment response, and the dose could be approximately 20–40 mg/kg/day, as has been previously reported [9,10,203]. Launching

phase I and II dose escalation clinical trials using high doses of aprepitant is needed to learn about drug-drug interactions, tolerability, safety, and efficacy. What if it is simply a question of dosage? [132,203] In the proposed studies, aprepitant can be administered alone (20–40 mg/kg/day for a long time) or in combination with chemotherapy or radiotherapy; aprepitant should be added to the chemotherapy protocol on each cycle on the same day as chemotherapy, or administered during the time of radiotherapy. In both cases, the dose should be 20 mg/kg/day [1]. This combination strategy is a promising approach to attain fewer sequelae and higher cure rates in patients with cancer. This result must be confirmed in future clinical trials. It is crucial to know the highest safe antitumor dose of aprepitant as well as its administration time. In summary, NK-1R is an antitumor therapeutic target, NK-1R antagonists are promising antitumor drugs, and much data supports the re-profiling of aprepitant as an antitumor drug. The re-profiling of aprepitant will be a rapid alternative for new antitumor strategies, considerably decreasing the risk for cancer patients. The USA National Comprehensive Cancer Network suggests that, for those patients showing a median overall survival of about 1 year and for whom the standard antitumor therapeutic strategies have failed, the best solution is their inclusion in clinical trials in which new antitumor strategies are tested. In this sense, the data reported here encourage the inclusion of these patients in clinical trials testing the antitumor effect of aprepitant, either alone or in combination therapies. Unfortunately, despite the numerous findings demonstrating the involvement of NK-1R in cancer, most of the current anticancer research is focused on other cancer research fields. In addition, pharmaceutical companies show, in general, no interest in the crucial roles played by the SP/NK-1R system in cancer. Table 2 describes some promising future research lines focused on the involvement of NK-1R in cancer.

**Table 2.** Future directions on NK-1R research in cancer: NK-1R as an antitumor target.

| |
|---|
| Single nucleotide variants (SNPs): their pathological significance has not been analyzed |
| To know the molecular mechanisms involved in NK-1R overexpression |
| To know how cancer cells express more truncated than full-length isoforms |
| No information on the formation of NK-1R dimers/oligomers has been provided: its capacity to heterodimerize is possible |
| To confirm whether the truncated form prolongs SP response |
| To know how the total number of full-length/truncated NK-1R isoforms is involved in cancer progression and the antitumor efficacy of NK-1R antagonists |
| To know the physiological significance of the presence of SP/full-length NK-1R form in the nucleus and the truncated NK-1R form in the cytoplasm |
| To confirm whether NK-1R regulates the balance between oxidant/antioxidant components of the redox system |
| To know why SP did not exert a proliferative action in specific tumor cells and how SP exerted an antimetastatic effect in some cases |
| The roles played by SP in the tumor microenvironment must be elucidated |
| To confirm whether the overexpression of NK-1R by cancer cells is a prognostic biomarker and whether an increased serum SP level is a predictive factor indicating a high risk of developing cancer or tumor development |
| Aprepitant administration before/after surgical procedures has been suggested to prevent recurrence and metastasis: this must be confirmed |
| To confirm whether patients with pancreatic ductal adenocarcinoma and high levels of NK-1R show a better overall survival |
| To know the function–structure relationships between NK-1R and SP for designing new antitumor drugs |

**Table 2.** *Cont.*

| |
|---|
| Crystal structures of NK-1R bound to antagonists are crucial for designing compounds with specific characteristics related to their binding and specificity properties; these studies must be developed |
| Strategies to increase the bioavailability of aprepitant must be performed |
| To know how NK-1R antagonists decrease the number of PD-1-positive cells |
| Aprepitant is a promising antitumor agent against rhabdoid tumors, either alone or in combination therapy (chemotherapy): this must be confirmed |
| Combination strategy (chemotherapy or radiotherapy + aprepitant): a promising approach for fewer sequelae and higher cure rates in patients with cancer: this must be confirmed |
| Phase I and II dose escalation clinical trials with high doses of aprepitant are needed to learn about drug-drug interactions, tolerability, safety, and efficacy: aprepitant can be administered alone (20–40 mg/kg/day for a long time) or in combination therapy with chemotherapy or radiotherapy |
| Antiemetic aprepitant drug: a crucial antitumor agent candidate targeting the NK-1R. Its repurposing as a new therapeutic strategy to overcome cancer is urgently needed |

**Author Contributions:** Conceptualization, R.C., F.D.R. and M.M.; resources, R.C. and F.D.R.; writing—original draft preparation: R.C. and F.D.R.; writing—review and editing, R.C., F.D.R. and M.M.; supervision, R.C. and F.D.R. All authors have read and agreed to the published version of the manuscript.

**Funding:** This research received no external funding.

**Informed Consent Statement:** Not applicable.

**Conflicts of Interest:** The authors declare no conflict of interest.

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
