# Peer review of "The Neurokinin-1 Receptor: A Promising Antitumor Target"

_2813-2564, doi:10.3390/receptors1010005_

Round 1

Reviewer 1 Report

This is a comprehensive review describing the role of SP/NK1R system in cancer and the possibility of using NK1 antagonists as anticancer agents. The manuscript is based on 205 mostly current citations and will be of interest for both chemists and pharmacologists.  The figures and Tables are well prepared and very informative. In my opinion this review can be published in the Receptors in its present form.

Reviewer 2 Report

This is a very comprehensive literature review about the implication of NK1R/SP in cancer. The topic is relevant and well supported by the references. Only minor suggestions are included:

1- Section 2 is too long. Consider breaking up into smaller sections with subheadings.

2- What are the challenges that NK1R-targeted cancer therapeutics would face? This review provides a lot of references to support the antagonism of NK1R as a cancer treatment but it does not provide much information of the limitations. I suggest limitations are discussed in a separate section, for example, are there specific cancers in which NK1R is not implicated?

3- The variants are not discussed in much depth in regard to the variability of treatment between individuals. What is the frequency of such SNPs? Perhaps the occurrence frequency should be added to the table.

4- How are the specific signaling pathways linked to cellular outcomes?

5- Line 14 NK-1R nomenclature is not consistent and reads NK 1receptor.

6- Line 71 reads single nucleotide variants (SNPs) but it should be polimorphisms or (SNVs).

Reviewer 3 Report

Substance P (SP) regulates multiple biological processes through its high-affinity neurokinin-1 receptor (NK-1R). The SP/NK-1R signaling axis is involved in the molecular bases of many human pathologies, including cancer. This review outlined the SP/NK-1R complex as a key player in human cancer by regulating the cancer cells proliferation, migration and metastasis. They evaluated NK-1R antagonists that may be useful in the development of new antitumor strategies. They also pointed out some future research lines focused on the involvement of NK-1R in cancer.

1. Figure 2 and Figure 4 seem familiar. They both represent NK-1R downstream pathways that are possibly involved and collaborate in cancer-associated processes. More details need to be illustrated, like the transcription factor or pathways involved.

2. What roles do SP/NK-1R complex played in physical and pathological situation?

3. They mentioned that truncated NK-1R isoforms could promote the malignant transformation of non-tumorigenic cells. The truncated form is an oncogenic isoform. Are there any references that explain the mechanism? How exactly does it affect tumor progression? Do the full-length/ truncated isoforms have different downstream pathways? 

4. NK-1R antagonists (e.g., Aprepitant, SR-140,333, CP-96,345, L-732,138, L-733,060) do exert an Antitumor effect. It is better to have a table list to describe the antitumor effect, mechanism, therapy strategies and clinical procedures for these main antagonists.

Round 2

Reviewer 3 Report

On page 11, lines 493-489, they mentioned there are five NK-1R antagonists approved for clinical practice. Then they gave six samples after this.

This manuscript has been sufficiently improved. It is considered to be accepted after modifying this typo.